# Antimicrobial Prescribing in the Emergency Department; Who Is Calling the Shots?

**DOI:** 10.3390/antibiotics10070843

**Published:** 2021-07-10

**Authors:** Laura M. Hamill, Julia Bonnett, Megan F. Baxter, Melina Kreutz, Kerina J. Denny, Gerben Keijzers

**Affiliations:** 1Department of Emergency Medicine, Christchurch Hospital, Canterbury DHB, Christchurch 8011, New Zealand; Laura.hamill@cdhb.health.nz; 2School of Medicine, Griffith University, Gold Coast, QLD 4215, Australia; Julia.Bonnett@griffithuni.edu.au (J.B.); megan.baxter@griffithuni.edu.au (M.F.B.); melina.kreutz@griffithuni.edu.au (M.K.); 3Department of Intensive Care, Gold Coast University Hospital, Gold Coast, QLD 4215, Australia; kerina.denny@health.qld.gov.au; 4Department of Emergency Medicine, Gold Coast University Hospital, Gold Coast, QLD 4215, Australia; 5Faculty of Health Sciences and Medicine, Bond University, Gold Coast, QLD 4215, Australia

**Keywords:** antibiotics, antimicrobial stewardship, appropriateness, emergency department, prescribing

## Abstract

**Objective:** Inappropriate antimicrobial prescribing in the emergency department (ED) can lead to poor outcomes. It is unknown how often the prescribing clinician is guided by others, and whether prescriber factors affect appropriateness of prescribing. This study aims to describe decision making, confidence in, and appropriateness of antimicrobial prescribing in the ED. **Methods:** Descriptive study in two Australian EDs using both questionnaire and medical record review. Participants were clinicians who prescribed antimicrobials to patients in the ED. Outcomes of interest were level of decision-making (self or directed), confidence in indication for prescribing and appropriateness (5-point Likert scale, 5 most confident). Appropriateness assessment of the prescribing event was by blinded review using the National Antibiotic Prescribing Survey appropriateness assessment tool. All analyses were descriptive. **Results:** Data on 88 prescribers were included, with 61% making prescribing decisions themselves. The 39% directed by other clinicians were primarily guided by more senior ED and surgical subspecialty clinicians. Confidence that antibiotics were indicated (Likert score: 4.20, 4.35 and 4.35) and appropriate (Likert score: 4.07, 4.23 and 4.29) was similar for juniors, mid-level and senior prescribers, respectively. Eighty-five percent of prescriptions were assessed as appropriate, with no differences in appropriateness by seniority, decision-making or confidence. **Conclusions:** Over one-third of prescribing was guided by senior ED clinicians or based on specialty advice, primarily surgical specialties. Prescriber confidence was high regardless of seniority or decision-maker. Overall appropriateness of prescribing was good, but with room for improvement. Future qualitative research may provide further insight into the intricacies of prescribing decision-making.

## 1. Introduction

Antimicrobial resistance (AMR) has been declared a global health crisis and if not tackled could cause up to 10 million deaths worldwide per year by 2050 [1]. While appropriate and timely antibiotic therapy saves lives [2,3,4] the misuse and overuse of antimicrobials accelerates the development of AMR and threatens our ability to treat infectious diseases, resulting in prolonged illness, disability, death, and increasing health care costs [5]. Overuse has harms on an individual level; from minor allergic reactions to potentially fatal infections, as well as serious medication interactions.

Errors in medication related problems are common, affecting 4–7% of medication orders [6,7]. Antimicrobial prescriptions in particular have shown greater incidence of prescription errors than other prescribing events, especially when prescribed by junior doctors. It has been estimated by the Institute of Medicine that medication errors cause 1 of 131 outpatient and 1 of 854 inpatient deaths [8]. Thus all clinicians should espouse caution and critical thinking when prescribing antimicrobials. 

The Emergency Department (ED) is a uniquely challenging environment for prescribers, with high-volume care, frequent interruptions, and competing priorities. ED clinicians frequently face diagnostic uncertainty and the threat of patient deterioration whilst awaiting results of investigations [9,10]. The challenge facing ED doctors is not only one of whether to prescribe antibiotics, but also of which type, route, frequency and dose, sometimes in the absence of a clear infective source or diagnosis. Ideally, clinicians are compliant with guidelines such as the surviving sepsis campaign [11] however, most patients with infection admitted from EDs do not have sepsis, yet are frequently commenced on broad-spectrum, parenteral antibiotics [12,13]. 

Most patients in hospital are admitted via the ED, consequently antimicrobial prescribing in EDs impacts the patterns of antimicrobial use across the hospital [14,15]. Additionally, prescriptions started in the ED are often continued in the community setting [14], impacting outpatient antimicrobial use. However, antimicrobial stewardship initiatives rarely focus on the ED [14]. A descriptive ED-based study in 2017 showed that in 33% of all patients receiving an antimicrobial prescription, the prescription was considered inappropriate [16]. This data needs further exploration as the antimicrobial prescribing clinicians may not be the primary decision makers and may have been directed by supervising clinicians, peers or inpatient specialty teams. To get further insight in how to remediate inappropriate prescribing in ED, we need to understand who is making the prescribing decisions in ED.

This study aims to describe factors associated with overall and appropriate antibiotic prescribing in the ED, including seniority, specialty of decision-makers and level of confidence of the prescriber, in both independent and guided prescribing decisions. The overarching goal of this work will be to inform future targeted steps towards improving antimicrobial stewardship (AMS) in the ED as well as to develop novel solutions.

## 2. Methods

### 2.1. Design, Setting and Participants

This was a descriptive study using both medical record review and questionnaires. The study was conducted in two EDs within the same health service in Queensland, Australia. The first ED serves a large tertiary level 750-bed hospital, and the second ED serves a 403-bed urban district hospital. In 2019 the two sites had a combined ED attendance of around 174,000 patients (112,000 and 62,000). Both hospitals are affiliated with local university and have medical and nursing students on placement. The combined staffing of the ED included doctors with a range of experience (63 consultants, 56 registrars, 81 residents and 19 interns), as well as nurse practitioners (8). Nurse practitioners have completed additional study at Master’s degree level and are senior and independent clinician with ability to prescribe antibiotics. Participants were a convenience sample of clinicians who prescribed antimicrobials to patients in the ED during data collection periods over two months in the winter of 2019. Within this health service all levels of medical practitioners can prescribe independently for all types of patients. Several guidelines, including the statewide sepsis pathway are available, which can prompt for senior medical officer input, but no formal policies or antimicrobial restrictions are enforced.

### 2.2. Questionnaire Design and Data Collection

The questionnaire questions were determined *a-priori* and pilot tested for face validity by three clinicians not part of the research team. After review by a qualitative research expert, questions were further refined. (Questionnaire; Appendix A).

Two research assistants collected data during six 12-h time periods (Table 1). A variety of shifts were selected including weekdays, weekends, days and nights, in order to minimize potential sample bias.

All patients who presented to the ED for the six pre-determined 12-h periods were identified. The two research assistants used the integrated electronic medical record (ieMR) to manually review the medication record of every patient. If the patient had been prescribed oral or parenteral antibiotics during their ED attendance, their prescribing doctor was eligible for participation and contacted to complete the questionnaire. Prescribing events were not eligible if the prescription was for topical medication, antivirals/antifungals, or the patient did not receive the prescribed medication whilst in the ED. The medical record was reviewed to determine prescribing doctor and indication for antibiotics as well some detail on clinical variables. All eligible clinicians were invited to complete the questionnaire within the same shift (in person) or at the latest 48 h after prescribing (in person or via telephone) as to minimize recall bias. Eligible clinicians were given the explicit option to decline participation.

When a clinician did not make their own decision, this was identified as a ‘guided’ or ‘directed’ decision. This direction, guidance or advice could relate to any, some or all components (type of drug, dose, route, duration) of the prescription. We defined clinicians as *senior* (consultants and registrars), *mid-level* (senior house officers and principal house officers) and *junior* (junior house officers and interns). Some prescribing decisions were made jointly (e.g., Senior ED clinician in consultation with mid-level inpatient team). For the purposes of analysis and simplicity, the most senior person noted was considered the primary decision maker. Confidence in prescribing was measured using a 5-point Likert scale with 5 representing high confidence.

### 2.3. Assessment of Appropriateness

Antibiotic appropriateness was independently assessed by two researchers who used the National Antibiotic Prescribing Survey (NAPS) tool [16,17] (Table A1), which has been used with a high rate of inter-rater reliability and validity [17]. As no gold standard for appropriateness exists, assessments were based on interpretation of clinical record review. Prescribing which deviated from guidelines could still be classed as appropriate if clear reasons were given (such as first line medication being out of stock). The validity of this approach has been further demonstrated by the consistency of findings from nationwide hospital point-prevalence studies [18]. Each assessor allocated the antibiotic prescription as being Optimal (1), Adequate (2), Suboptimal (3), Inadequate (4), or Not Assessable (5) as per NAPS guidelines [17]. If there was no agreement, a senior researcher arbitrated the appropriateness. Ratings of 1 or 2 were given a final classification of appropriate and ratings of 3 or 4 were classified as inappropriate. 

### 2.4. Data Analysis

As this is a descriptive study, no formal power calculation was performed. All data is descriptive in nature. We described the prescribers as follows; firstly, by dividing them by independent decision-makers and directed decision makers, secondly by seniority.

Ethical approval was granted by the institutional Human Research Ethics Committee (HREC/17/QGC/41). Findings are reported in accordance with the Strengthening the Reporting of Observational studies in Epidemiology (STROBE) Statement for cohort studies [19].

## 3. Results

The questionnaire was distributed to 128 clinicians and returned by 94 (73% response rate) of those 88 met eligibility criteria (Figure 1). 

There were few missing data (less than 3% for any variable.) Seniority of the participants are summarised in Table 2, with over three-quarters of participants classed as mid-level or senior, with the most experienced respondent practicing for 25 years. Twenty-two respondents were emergency medicine trainees. The conditions for which antibiotics were prescribed are outlined in Table A2. The three most common indications were respiratory tract infections (30%), skin and soft tissue infections (20%) and urinary tract infections (17%).

### 3.1. Seniority and Specialty of Prescribing Decision-Making

Almost two-thirds (61%, 54/88) of participants made the prescribing decision themselves. Of the 39% (*n* = 34) of clinicians who did not make their own prescribing decision, 88% (*n* = 30/34) reported that a senior (consultant or registrar) clinician had guided them. There were 2 interactions whereby mid-level staff advising junior or mid-level staff in their prescribing and one instance of a nurse practitioner guiding mid-level staff. No junior specialty doctor(s) advised a senior on prescribing.

In cases where the participant was not the decision maker, over half (62%) were directed by a non-ED clinician. The specialty teams involved are shown in Figure 2. Inpatient specialties were often involved when the patient had previously been under their care as an inpatient, with known microbiological sensitivities available.

There were three cases where the infectious disease (ID) team was involved (3.4%). A surgical subspecialty registrar consulted ID for one patient and for the other two patients an existing ID antimicrobial plan was documented in the medical record. This plan was followed by the treating clinicians.

### 3.2. Resource Use

Three-quarters (74%, 40/54) of respondents who made their own decision indicated they used the national resource “Electronic Therapeutic Guidelines”^TM^ to aid their decision making [20]. Of those who indicated they made their own decision and used eTG (35/40) 87.5% were appropriate and followed guideline recommendations. Eleven (12.5%) of respondents used other guidelines. Five (5.7%) of respondents quoted “ED experience” as the resource that they used. A quarter (23.9%) of respondents used multiple resources (*n* = 21) and 6.8% (*n* = 6) used an ED senior as a resource for direct advice. 

### 3.3. Prescribing Confidence by Decision-Maker and Seniority

Confidence in prescribing was high across all groups. If the prescribing clinician was not the decision maker, they were less frequently ‘very confident’ that antibiotics were indicated or appropriate (Figure 3a,b). Confidence was similar across all groups regardless of seniority. Mean confidence on a Likert scale from 1–5 that antibiotics were indicated (4.20, 4.35 and 4.35, respectively) and appropriate (4.07, 4.23 and 4.29, respectively) was similar for juniors, mid-level and senior prescribers.

### 3.4. Appropriateness of Prescribing

Eighty-five percent (75/88) of prescribing was assessed as appropriate using the NAPS tool. Proportions of appropriate prescribing were similar, when comparing by seniority, independent vs directed prescribing or different levels of confidence (Table 3).

## 4. Discussion

This study describes the novel concept of decision-making in antibiotic prescription, in contrast with most studies which focus on the act of prescribing. We found that nearly two-thirds of clinicians who prescribe antibiotic medication decide this themselves, mostly with the support of endorsed guidelines. Over one-third of prescribing was guided or directed by senior ED clinicians or subspecialty advice. Prescriber confidence that antibiotics were indicated or appropriate was high, regardless of seniority or whether prescribing was self-directed or directed by others. Overall appropriateness of prescribing in this study was 85%, with similar proportions of appropriate prescribing when comparing seniority, decision-maker and prescriber confidence.

### 4.1. Seniority and Specialty of Decision-Making

Our study shows that mid-level and senior ED clinicians conduct most prescribing, with the junior cohort responsible for less than 20% of prescriptions. This is reassuring, as independent decision-making skills are still developing in this junior group. This is in contrast with a recent paper which showed that seventy percent of hospital prescribing is done by doctors in their first two years after medical school [21]. This difference may be explained by the difference in setting, where in the ED there is usually a senior clinician available for direct consultation.

Of particular interest was the decision-making process for patients who were geographically in the ED and were to be admitted under an inpatient specialty. In nearly 60%, when participants had prescribed an antibiotic and the primary decision maker was a non-ED clinician, this was directed by surgical specialties. Uncertainty about admission, institutional hierarchy, perceived urgency, and possible delays to specialty review all can influence prescribing decisions [22]. It is crucial to patient safety that there is a collaborative approach to patient care, including antimicrobial decisions, as there is an important trade-off between timely and inappropriate prescribing [21,22].

In our study, surprisingly no ED clinicians consulted with the infectious disease team. This may be due to the time constraints, but also may represent a knowledge gap on when to consult appropriately. It is possible that prescribing clinicians feel that this is core business, and they should be able to prescribe without specialist advice. The finding that 1 in 6 patients received inappropriate antibiotics suggest a more robust approach may be required. Also, after study design but before data collection, our health service ceased to have an on-call microbiology registrar, limiting consultation options.

### 4.2. Prescribing Details

Of note, clinicians often felt the need to further explain their decision-making process by hand-written notes on the paper questionnaire. In nine cases (10%) the clinicians stated they had deviated from guidelines due a nation-wide benzylpenicillin shortage. This demonstrates the complexities inherent in having a one-size-fits-all approach to antibiotic prescribing; nuance is required on a patient level. This is a barrier to effective guidelines, as patients will invariably have idiosyncrasies which lead to deviation. However, with developing machine learning and artificial intelligence, perhaps individualised recommendations may be the future of AMS. Decision support tools and smartphone apps have shown value in this space [23,24].

### 4.3. Prescribing Confidence

In our study confidence in own prescribing was high, with similar level of confidence in senior doctors and juniors, although respondents tended to be more confident in their own decisions than decision of others. Prior studies indicate that medical students and junior doctors have important shortcomings in the domain of prescribing, especially with respect to antimicrobials [25]. Despite this known weakness, there is little targeted teaching around antimicrobial choice in early clinical practice. Some junior clinicians maybe exhibiting the “unconscious incompetence” [26] of early clinical practice, which has been previously described [27,28]. However in our study appropriateness was similar for junior and senior staff.

### 4.4. Appropriateness

In our sample, 85% of prescriptions were deemed appropriate (NAPS 1 or 2) which compares favourably to the 67% appropriateness found in the same setting using the same methodology in 2017 [16]. This difference may be partially explained by chance (as our study was small), improved practices in our setting, or because of our design which allowed clinicians to clarify any decisions that were not strictly following guidelines. It highlights that non-compliance with guidelines cannot be entirely interpreted as inappropriate prescribing *per se*, although this finding warrants further study. Our study found similar appropriate prescribing by seniority, decision making or confidence and a larger study focusing on these factors would be required.

### 4.5. Limitations

This was a dual site study in a single season and findings may not be generalisable to other settings. Despite a good response rate of 73% and limited missing data, we cannot exclude selection bias. Although, we believe that as only one eligible respondent declined participation it is unlikely to have had a significant impact. Consultants only represented 3% of participants, but this is consistent with prior prescribing studies. Further, our study has limitations common to all survey-based research. Our questionnaire was purpose-made and although tested for face-validity it had limited answer options, making interpretation of more nuanced clinical situations challenging. This is highlighted by respondents using free text comments to further explain their decisions. The cross-sectional nature of this study provides a static picture of a dynamic process. This is further limited by using the terminology ‘guided’ or ‘directed’ prescribing. We acknowledge that by using this terminology certain subtleties in the decision making and prescribing etiquette cannot be commented on [29]. Furthermore, we cannot comment on what effect ED nursing or pharmacy staff may have had on prescribing habits. Response bias or recall bias may have led to erroneous or misleading answers, this was mitigated by only allowing answers up to 48 h post prescribing. Participants on night shift contributed to our non-response rates. Lastly, clinicians may be reluctant to admit that they did not use endorsed guidelines, as only two respondents stated they did not use guidelines. Given the busy nature of emergency medicine, this proportion is likely to be higher.

### 4.6. Recommendations

Our study has provided further insight in antimicrobial prescribing, decision-making and confidence. It provides useful information to inform future work related to individualised prescribing. Such prescribing will place the patient at the center of the decision making with a focus on areas where inappropriate prescribing is currently most common. Future qualitative research will be required to provide further insight into the intricacies of prescribing decision-making.

## 5. Conclusions

Nearly two-thirds of ED clinicians who prescribe antibiotic medication decide this themselves, usually supported by guidelines. In over one-third of prescribing was directed by senior ED clinicians or based on specialty advice, primarily surgical specialties. Prescriber confidence was high regardless of seniority or decision-maker. Overall appropriateness of prescribing was good with further room for improvement.

## Figures and Tables

**Figure 1 antibiotics-10-00843-f001:**
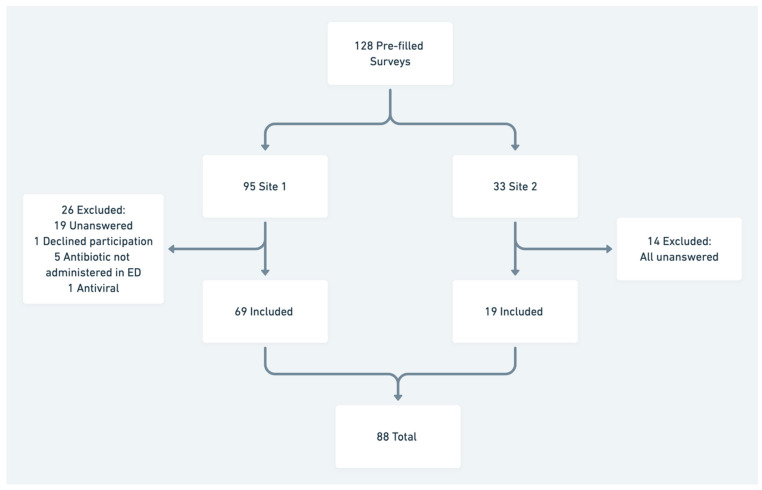
Flow chart of included participants.

**Figure 2 antibiotics-10-00843-f002:**
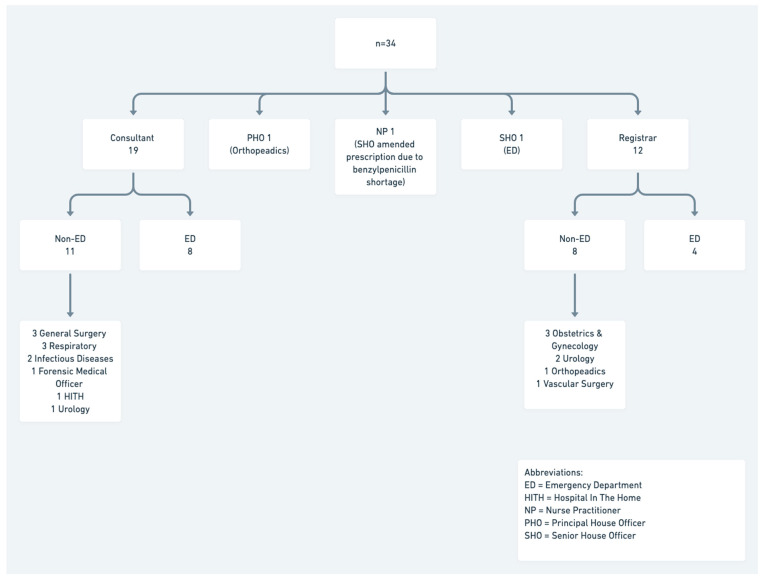
Antibiotic prescribing guided by other staff than prescriber (*n* = 34).

**Figure 3 antibiotics-10-00843-f003:**
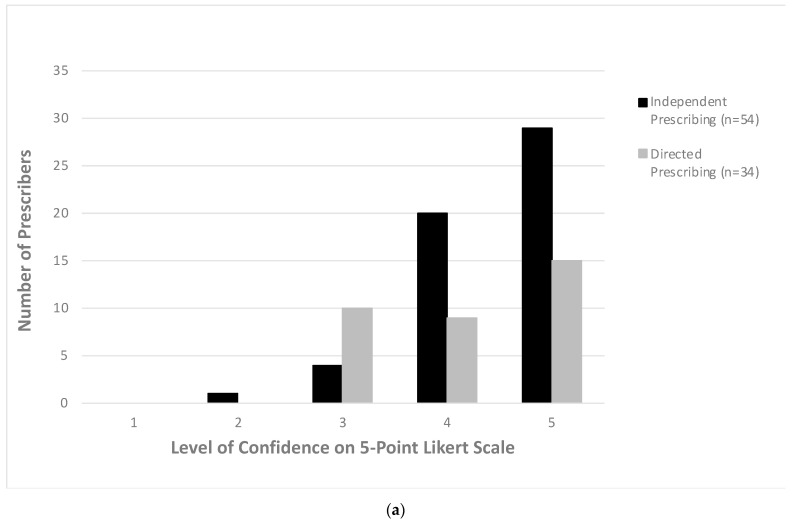
(**a**) Prescriber confidence if antibiotic prescription was indicated. (**b**) Prescriber confidence if antibiotic prescription was appropriate.

**Table 1 antibiotics-10-00843-t001:** Data Collection Periods.

Site 1	Site 2
23/08/19 20:00–08:00 (Friday–Saturday)	17/09/19 08:00–20:00 (Tuesday)
26/08/19 08:00–20:00 (Monday)	28/09/19 08:00–20:00 (Saturday)
12/09/19 08:00–20:00 (Thursday)	29/09/19 20:00–08:00 (Sunday–Monday)

**Table 2 antibiotics-10-00843-t002:** Characteristics of Respondents.

Seniority Classification	*n* (%)	Job Role	*n* (%)	Specialty—*n* (%)
Senior	31 (35.2)	Consultant	3 (3.4)	Emergency—28 (90.3) Respiratory—2 (6.5) Urology—1 (3.2)
		Registrar	28 (31.8)	
Mid-level	38 (43.2)	Senior House Officer	31 (35.2)	Emergency—35 (92.1) Obstetrics & Gynaecology—3 (7.9)
		Principal House Officer	7 (8.0)	
Junior	14 (15.9)	Junior House Officer	4 (4.5)	Emergency—13 (92.9) Orthopaedics—1 (7.1)
		Intern	10 (11.4)	
Other	5 (5.7)	Nurse Practitioner	4 (4.5)	
		Unknown	1 (1.1)	

**Table 3 antibiotics-10-00843-t003:** Appropriateness of Prescribing.

	Seniority.	*n* (%)	Appropriate * *n* (%)
Seniority of Respondents	Senior	31 (35.2)	26 (84)
	Mid-level	38 (43.2)	33 (87)
	Junior	14 (15.9)	12 (86)
	Other	4 (4.5)	3 (75)
	Unknown	1 (1.1)	1 (100)
Seniority of decision maker	Senior	51 (60.0)	44 (86)
	Mid-level	28 (31.8)	24 (86)
	Junior	3 (3.4)	3 (100)
	Other	5 (5.7)	4 (80)
	Unknow	1 (1.1)	1 (100)
Decision	Independent	54 (61.4)	47 (87)
	Directed	34 (38.6)	28 (82)
Confidence level Antibiotic Indicated—Independent	4 or 5	49 (91)	43 (88)
	1, 2 or 3	5 (9)	4 (80)
Confidence level Antibiotic Indicated—Directed	4 or 5	24 (71)	19 (79)
	1, 2 or 3	10 (29)	9 (90)
Confidence level Antibiotic Appropriate—Independent	4 or 5	50 (93)	44 (88)
	1, 2 or 3	4 (7)	3 (75)
Confidence level Antibiotic Appropriate—Directed	4 or 5	26 (76)	21 (81)
	1, 2 or 3	8 (24)	7 (88)

***** Appropriateness assessed by NAPS assessment [17].

## Data Availability

The data presented in this study are available on request from the corresponding author. The data are not publicly available as this was not covered by our ethics approval.

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
