# Peer review of "Antimicrobial Prescribing in the Emergency Department; Who Is Calling the Shots?"

_antibiotics, 2021, doi:10.3390/antibiotics10070843_

Round 1

Reviewer 1 Report

Hamill et al. have presented an interesting antibiotic stewardship assessment detailing the Australian point of view. I mainly have two comments.

  • Both in the abstract and in the text body of the manuscript, the conclusions mainly repeat the results. Nothing else to learn or to conclude from the data? Any suggestions for follow-up-studies? Or consequences for the future management? The authors mention “room for improvement”: What exactly do they want to improve and how?
  • Methods, “Assessment of appropriateness”: Are there any information on sensitivity and specificity of the applied assessment scheme? How exactly did the authors take deliberate deviations from guidelines into account, which are sometimes necessary for good reason? Can 95%-confidence intervals of sensitivity and specificity of this measurement approach account for the differences compared to the 2017-assessment?

Author Response

Comment 1

Both in the abstract and in the text body of the manuscript, the conclusions mainly repeat the results. Nothing else to learn or to conclude from the data? Any suggestions for follow-up-studies? Or consequences for the future management? The authors mention “room for improvement”: What exactly do they want to improve and how?

Response 1

The reviewer wonders if we can have stronger conclusions including steps for future research. As this was a descriptive study with a modest sample size outlining current real-world practice, we did not want to overstate the findings. As such we chose a modest conclusion based on the data. Follow-up research is indeed being planned, and we have now referred to recommendations or future (qualitative) work in both the abstract conclusion as well as the Discussion, to highlight the need to explore the subtleties of decision making.

With regards to the comment regarding ‘room for improvement’, this refers to appropriate prescribing. Ideally all antibiotics prescriptions are appropriate. To improve appropriateness of prescribing, a suite of interventions may be required. This could include electronic support, nudge initiatives or force functions (eg. Infectious Disease approval prior to being able to prescribe certain antibiotics, before prescribing can occur). These ideas are all fertile ground for future work but are outside the scope of this work.

Comment 2

Methods, “Assessment of appropriateness”: Are there any information on sensitivity and specificity of the applied assessment scheme? How exactly did the authors take deliberate deviations from guidelines into account, which are sometimes necessary for good reason? Can 95%-confidence intervals of sensitivity and specificity of this measurement approach account for the differences compared to the 2017-assessment?

Response 2

Appropriateness was measured using the National Antimicrobial Prescribing Survey tool. This is a tool used in prior research, but no test accuracy (sensitivity/specificity) results are reported in the literature. This is (likely) due to the fact that it is difficult to establish a ‘gold standard’ for appropriateness to compare against (A ‘true positive’ likely still requires a panel, or expert consensus). The NAPS is used by independent assessors and includes assessment of all clinical data to categorise appropriateness. This information may include reasons why there was deviation from guidelines and can be incorporated in the assessment. If there was disagreement a 3rd person arbitrated. One of the features of the NAPS tool is that if a decision is made based on a senior Infection Disease consult, it automatically is judged as an appropriate prescription.

In summary, we agree there is potential for variation of interpretation, however our approach is consistent with prior literature.

We have added a sentence in the methods to clarify more detail on the NAPS tool

Reviewer 2 Report

The study authors used a combination of a questionnaire and medical record review to describe the level of independence of antibiotic decision-making and confidence about these decisions in the emergency department setting. Given that many antibiotics are initiated in the ED and antimicrobial stewardship initiatives are challenging in this setting, this study could provide important insight into opportunities to improve antibiotic prescribing. While the paper is well-written, the antibiotic decision making process is essentially reduced to "directed" vs "non-directed" which may occlude much of the nuance associated with the interpersonal aspects of these decisions. Perhaps semi-structured interviews may help to acquire additional detail about this process. Suggestions for improvement are included below:

ABSTRACT
1. Were the two EDs included in the study part of academic teaching hospitals, if so how many of the participants were trainees?
2. Were non-physicians who prescribed antimicrobial agents included?

INTRODUCTION
3. It seems that the term "directed by" to describe prescribing decisions may lack the nuance that occurs in hospitals where others may "influence" prescribing decisions more subtly due to hierarchy rather than explicitly telling other clinicians what to do. Consider citing work by Esmita Charani and colleagues on "prescribing etiquette":
https://doi.org/10.1093/cid/cit212

METHODS
4. Lines 82-84: are nurse practitioners considered "doctors" in Australia? If not please consider revision to this sentence, or simply a close bracket is missing.
5. Since antifungals and antivirals were excluded, it would seem that using the term "antibiotics" throughout rather than "antimicrobials" would be more appropriate.
6. "For the purposes of analysis and simplicity, the most senior person noted was considered the primary decision maker." Since the main objective of this work is to describe prescribing decision making, I am concerned that this assumption may obscure the actual influence and interpersonal dynamic that occurs in the ED. Further rationale may be required to support this assumption.
7. Non-physician clinicians may also provide an influence on prescribing decisions as well (e.g., pharmacists, nurses). Was this addressed?
8. A definition for being "directed" or making the "decision" to prescribe may be needed. Is this in a formal sense, or could it include something less formal like a suggestion? What aspects of the prescription could be directed, e.g., decision to prescribe vs. the selection of antibiotic/dose vs. the duration?
9. Please consider a couple of sentences describing the state of antimicrobial stewardship in the study institutions, were there educational opportunities, audit and feedback, any antimicrobial restrictions, etc?

RESULTS
8. Did any participant prescribe more than one antibiotic prescription during the study period, if so, how was that dealt with (was only 1 prescription included/assessed)?
9. Line 172, how does using an "ED senior as a resource" differ from being directed to prescribe?
10. Line 174 to 175 indicates that prescribers who were not the decision maker, they were "less likely to be very confident that antibiotics were indicated or appropriate". However since this study is descriptive and no inferential statistics were performed it may be difficult to make this statement with confidence. Perhaps adding the term "numerically" to acknowledge this was not assessed for statistical significance, or include inferential stats to support the statement.

DISCUSSION
11. Similarly, line 241 "However in our study we did not find a difference in appropriateness by seniority." and line 250-252 "Our study did not detect any differences in appropriate prescribing by seniority, decision making or confidence". Caution should be used in these statements given there were no statistical tests to assess for these difference and a sample size calculation was not performed.

APPENDIX
12. There are two minor errors that require correction: a. C. difficile should have a small "d" and be italicized and b. Pneumonia is spelled incorrectly.

Author Response

Comment 1

ABSTRACT
1. Were the two EDs included in the study part of academic teaching hospitals, if so how many of the participants were trainees?

Response 1

The two hospitals are part of the same health service and training network with staff rotating over both EDs. Indeed, the main campus is a tertiary hospital with the second campus also having a teaching role. Both campuses have medical and nursing students from two different universities. We have clarified this in the methods. Twenty-two respondents (about a quarter) were trainees, we have added this to the results.

  1. Were non-physicians who prescribed antimicrobial agents included?

Response 2

Yes, nurse practitioners are included. Nearly all prescribing in our setting is by medical doctors although nurse practitioners (NPs) have ability to prescribe, as evidenced by 4 NPs being included as participants (Table 1). We have clarified the role of NPs in the methods.

INTRODUCTION
3. It seems that the term "directed by" to describe prescribing decisions may lack the nuance that occurs in hospitals where others may "influence" prescribing decisions more subtly due to hierarchy rather than explicitly telling other clinicians what to do. Consider citing work by Esmita Charani and colleagues on "prescribing etiquette":
https://doi.org/10.1093/cid/cit212

Response 3

Thank you for this comment and the reference. Charani et al report: “Senior doctors consider themselves exempt from following policy and practice within a culture of perceived autonomous decision making that relies more on personal knowledge and experience than formal policy.”

We agree that many factors may influence decision making, and we agree that the term ‘directed’ could be interpreted as fairly formal and hierarchical. As a compromise, we have chosen the term ‘guided’ as this probably strikes the balance.  We have added a sentence in the methods and have changed ‘directed’ to ‘guided’ in most instances, and we have added the need for considering prescribing etiquette, including the reference, in the limitations section of the discussion.

METHODS
4. Lines 82-84: are nurse practitioners considered "doctors" in Australia? If not please consider revision to this sentence, or simply a close bracket is missing.

Response 4 –

Thank you for pointing this out. Indeed, nurse practitioners are not doctors. We have added a close bracket.

  1. Since antifungals and antivirals were excluded, it would seem that using the term "antibiotics" throughout rather than "antimicrobials" would be more appropriate.

Response 5 –

Thanks for pointing this out. This study is about antibiotic prescribing which is part of overall antimicrobial stewardship. We have adjusted where required.

  1. "For the purposes of analysis and simplicity, the most senior person noted was considered the primary decision maker." Since the main objective of this work is to describe prescribing decision making, I am concerned that this assumption may obscure the actual influence and interpersonal dynamic that occurs in the ED. Further rationale may be required to support this assumption.

Response 6 –

This line is in response to a small number (less than 5) of guided-prescribing surveys where the respondent noted down both a mid-level and senior level clinician as helping with their decision. We acknowledge that our study design did not allow to interrogate the subtleties of decision-making. In our experience, it would be unlikely for a junior to ask for advice and then completely ignore that advice.

We have added this as a limitation in the Discussion section.

  1. Non-physician clinicians may also provide an influence on prescribing decisions as well (e.g., pharmacists, nurses). Was this addressed?

Response 7 –

In line with the response above, we were unable to comment on this due to our study design. One of our medical doctors received advice from a NP as currently in the manuscript, but we agree that the wider nursing cohort and ED pharmacists may have had an impact. Unfortunately, we cannot comment on the direction or effect size of their impact. We have expanded on this in the Discussion section.

  1. A definition for being "directed" or making the "decision" to prescribe may be needed. Is this in a formal sense, or could it include something less formal like a suggestion? What aspects of the prescription could be directed, e.g., decision to prescribe vs. the selection of antibiotic/dose vs. the duration?

Response 8 –

We appreciate the reviewer’s comment. As outlined, ‘directed’ sounds definitive and hierarchical. However, we see ‘directions’ more like asking for guidance (for example when you are lost in a new city. In our view asking for directions or being guided is the same as asking for advice). This advice could relate to any part of prescribing (drug type, dose, route, duration) although we did not ask this detail. We have added our definition of ‘directed’ to the methods.

  1. Please consider a couple of sentences describing the state of antimicrobial stewardship in the study institutions, were there educational opportunities, audit and feedback, any antimicrobial restrictions, etc?

Response 9 –

Currently the methods states: ”Within this health service all levels of medical practitioners can prescribe independently for all types of patients. Several guidelines, including the state-wide sepsis pathway are available, which can prompt for senior medical officer input, but no formal policies are enforced”

We have added that ‘there are no enforced antimicrobial restrictions’

RESULTS
10. Did any participant prescribe more than one antibiotic prescription during the study period, if so, how was that dealt with (was only 1 prescription included/assessed)?

Response 10 –

Indeed, there were 34 patients who received more than antibiotic prescription. These universally were for the same condition, for example for a pyelonephritis both ampicillin and gentamycin may have been prescribed. 

As the decision to prescribe is based on the condition/indication and advice/guidance usually included both medications, such prescriptions for a single condition were treated as one survey response.

  1. Line 172, how does using an "ED senior as a resource" differ from being directed to prescribe?

Response 11 –

The clinician deciding “to prescribe” indicate they were the primary decision maker to give antibiotics, then they used ED senior to help them work out the details of drug class, dose and route. This is the distinction from direct advice from the ED senior, versus them referring to a local guideline to adhere that the junior still had to look up. We have added the words ‘direct advice’

  1. Line 174 to 175 indicates that prescribers who were not the decision maker, they were "less likely to be very confident that antibiotics were indicated or appropriate". However, since this study is descriptive, and no inferential statistics were performed it may be difficult to make this statement with confidence. Perhaps adding the term "numerically" to acknowledge this was not assessed for statistical significance or include inferential stats to support the statement.

Response 12 –

Thank you, we agree and have re-worded this

DISCUSSION
13. Similarly, line 241 "However in our study we did not find a difference in appropriateness by seniority." and line 250-252 "Our study did not detect any differences in appropriate prescribing by seniority, decision making or confidence". Caution should be used in these statements given there were no statistical tests to assess for these difference and a sample size calculation was not performed.

Response 13 –

Thank you, we agree and have re-worded this throughout

APPENDIX
14. There are two minor errors that require correction: a. C. difficile should have a small "d" and be italicized and b. Pneumonia is spelled incorrectly.

Response 14 –

Thank you, we appreciate the attention tp detail and we have now corrected these errors.

Reviewer 3 Report

I read with great interest the paper. I find it well wrote and with good idea research. Only some minor suggestions but congratulation to the authors for the paper. 

Introduction: add if you have data on AMR in your country and describe how AMR is a global health problems also for African countries (see Maternal caesarean section infection (MACSI) in Sierra Leone: a case-control study.) and for these reason all countries are affected from AMR with high burden of mortality and disability

Methods and results: very clear

Discussion : add the role of medical education to improve awareness and knowledge in young doctor and the institution of course in AMR during medical degree (see and cite if you want Italian young doctors' knowledge, attitudes and practices on antibiotic use and resistance: A national cross-sectional survey. J Glob Antimicrob Resist. 2020 Dec;23:167-173.)

Give in conclusion section some public/global health proposal that came from your very interesting paper

Author Response

Comment 1:

Introduction: add if you have data on AMR in your country and describe how AMR is a global health problems also for African countries (see Maternal caesarean section infection (MACSI) in Sierra Leone: a case-control study.) and for these reason all countries are affected from AMR with high burden of mortality and disability

Response 1 –

As Australia is vast continent and our study is not representative of the whole country, we have elected not add any country specific data. The introduction highlights the global importance, as the introduction starts with: Antimicrobial resistance (AMR) has been declared a global health crisis….

Comment 2

Discussion : add the role of medical education to improve awareness and knowledge in young doctors and the institution of course in AMR during medical degree (see and cite if you want Italian young doctors' knowledge, attitudes and practices on antibiotic use and resistance: A national cross-sectional survey. J Glob Antimicrob Resist. 2020 Dec;23:167-173.)

Give in conclusion section some public/global health proposal that came from your very interesting paper

Response 2 –

We agree that a multifaceted suite of interventions will be required to create change, and this will include medical education, starting during the medical degree. We have proposed next steps for qualitative research in the Discussion.

Round 2

Reviewer 2 Report

Thank you for making these changes. I have no further suggestions.